# Rapid Generation of Long Noncoding RNA Knockout Mice Using CRISPR/Cas9 Technology

**DOI:** 10.3390/ncrna5010012

**Published:** 2019-01-23

**Authors:** Nils R. Hansmeier, Pia J. M. Widdershooven, Sajjad Khani, Jan-Wilhelm Kornfeld

**Affiliations:** 1Max Planck Institute for Metabolism Research, Gleueler Strasse 50, 50931 Cologne, Germany; Nils.Hansmeier@sf.mpg.de (N.R.H.); PWiddershooven@age.mpg.de (P.J.M.W.); Sajjad.Khani@sf.mpg.de (S.K.); 2Cologne Cluster of Excellence: Cellular Stress Responses in Ageing-associated Diseases, Medical Faculty, University of Cologne, Joseph-Stelzmann-Str. 26, 50931 Cologne, Germany; 3Institute for Prophylaxis and Epidemiology of Cardiovascular Diseases (IPEK), Ludwig Maximilian University of Munich, 80336 Munich, Germany; 4Department for Biochemistry and Molecular Biology, Functional Genomics and Metabolism Unit, University of Southern Denmark, Campusvej 55, 5230 Odense M, Denmark

**Keywords:** long noncoding RNA, clustered regularly interspaced short palindromic repeats (CRISPR)/Cas9-mediated genome engineering, knockout mice

## Abstract

In recent years, long noncoding RNAs (lncRNAs) have emerged as multifaceted regulators of gene expression, controlling key developmental and disease pathogenesis processes. However, due to the paucity of lncRNA loss-of-function mouse models, key questions regarding the involvement of lncRNAs in organism homeostasis and (patho)-physiology remain difficult to address experimentally in vivo. The clustered regularly interspaced short palindromic repeats (CRISPR)/Cas9 platform provides a powerful genome-editing tool and has been successfully applied across model organisms to facilitate targeted genetic mutations, including *Caenorhabditis elegans*, *Drosophila melanogaster*, *Danio rerio* and *Mus musculus*. However, just a few lncRNA-deficient mouse lines have been created using CRISPR/Cas9-mediated genome engineering, presumably due to the need for lncRNA-specific gene targeting strategies considering the absence of open-reading frames in these loci. Here, we describe a step-wise procedure for the generation and validation of lncRNA loss-of-function mouse models using CRISPR/Cas9-mediated genome engineering. In a proof-of-principle approach, we generated mice deficient for the liver-enriched lncRNA *Gm15441*, which we found downregulated during development of metabolic disease and induced during the feeding/fasting transition. Further, we discuss guidelines for the selection of lncRNA targets and provide protocols for in vitro single guide RNA (sgRNA) validation, assessment of in vivo gene-targeting efficiency and knockout confirmation. The procedure from target selection to validation of lncRNA knockout mouse lines can be completed in 18–20 weeks, of which <10 days hands-on working time is required.

## 1. Introduction

In recent decades, a rather unexpected finding from Next-Generation RNA Sequencing (RNA-Seq) initiatives such as ENCODE (encyclopedia of DNA elements) [1], FANTOM (functional annotation of the mammalian genome) [2] and NONCODE (an integrated knowledge database dedicated to ncRNAs) [3] was the observation that, whilst only two percent of genomes in higher organisms encode for protein-coding genes, more than two-thirds are transcribed across developmental stages and cell types [4]. Subsequently, this discovery led to the identification of several thousand so-called long noncoding RNAs (lncRNAs) [5,6] in mice and humans [7]. Although ascribing global and overarching molecular functions to all lncRNAs remains challenging, individual lncRNAs were shown to control a wide spectrum of cellular and molecular processes, ranging from microRNA sequestration [8], chromatin modifier recruitment [9,10], to partaking in higher-order circular RNA (circRNA)–lncRNA–microRNA regulatory circuits [11].

Historically, many lncRNA loss-of-function studies were performed using RNA interference (RNAi) in vitro where small interfering RNAs (siRNAs) or short hairpin RNAs (shRNA) were delivered into immortalized cell systems [12], yet validation of these in vitro findings, particularly using genetic loss-of-function approaches in vivo, is sparse [13]. To understand the molecular contribution of lncRNAs to health and disease, for instance by investigating the phenotypic consequences of lncRNA alteration in vivo, the field needs simple, robust, cheap and standardized approaches to disrupt lncRNA function and bypass some confounding factors of performing RNA interference [14].

The CRISPR/Cas9 (Clustered Regularly Interspaced Short Palindromic Repeats/CRISPR-Associated Protein 9) system has endowed us with a versatile platform, where the endonuclease Cas9 is recruited to specific genomic sites by virtue of a sequence-dependent CRISPR RNA (crRNA) and a sequence-independent trans-activating CRISPR RNA (tracrRNA) [15,16]—two RNA molecules that can be fused to a so-called single guide RNA (sgRNA) for simplicity reasons [17]. Upon sequence-specific, sgRNA-mediated Cas9 recruitment, DNA double-strand breaks (DSBs) are introduced that are subsequently repaired via error-prone non-homologous end-joining (NHEJ) or, if donor templates are provided, by homology-directed repair (HDR) [17]. Whereas NHEJ often results in random insertions and deletions of single nucleotides (indels) at cleavage sites, leading to gene-inactivating mutations, gene targeting via HDR can be used to precisely generate complex alleles [18].

Due to the broad applicability of the CRISPR/Cas9 system for gene-editing, we here present a detailed protocol for the generation of lncRNA knockout mice, including considerations for lncRNA gene targeting, workflows for sgRNA prediction and in vitro validation, CRISPR/Cas9 transgenesis via pronuclear micro-injection (PNI) of sgRNA–Cas9 complexes [19] and validation of lncRNA-deficient animals.

Classical gene targeting approaches using embryonic stem (ES) cells involve cloning of the targeting construct, rare homologous recombination (HR) events in ES cells and extensive ES cell culture techniques, for instance ES cell electroporation, expansion and selection. These procedures can take up to several months to generate a correctly targeted ES cell clone and ultimately a homozygous transgenic mouse line, whereas our methodology can generate loss-of-function mouse models within 18–20 weeks.

## 2. Results

### 2.1. Overview of the Protocol

Following this protocol, one can validate lncRNA targeting efficiency in vitro within 2 weeks, and can successfully generate lncRNA-deficient mouse lines within 4 months. This protocol is broken up into four parts, including lncRNA selection (Step 1), validation of sgRNA efficacy in vitro (Step 2), sgRNA–Cas9 ribonucleoprotein (RNP) assembly and pronuclear microinjection (Step 3), and validation of lncRNA-null founder animals followed by breeding with C57BL/6 mice for germline transmission (Step 4). In a proof-of-principle approach, we ablated the expression of lncRNA *Gm15441*, whilst leaving the expression of the antisense overlapping protein-coding *Txnip* gene intact.

#### 2.1.1. General Considerations to Design Long Noncoding RNA Gene-Targeting Strategies

In contrast to protein-coding genes, gene-targeting strategies using CRISPR/Cas9 mutagenesis for inducing nucleotide insertions/deletions (indels) are not applicable for most lncRNA genes: Frame-shift mutations caused by NHEJ-mediated repair of DSBs disrupt the reading frame of translation products, but remain ineffective for lncRNA genes due to their noncoding nature and the absence of long open-reading frames (ORFs). Those lncRNAs that execute their function mainly by inducing local transcription will not be affected by mutations of the genomic sequence, unless regulatory elements driving transcription of the lncRNA gene are also disrupted. However, regulatory regions remain uncharacterized for most lncRNAs and preclude specific targeting of most lncRNA genes.

Several groups have employed HDR of CRISPR/Cas9-induced DSBs to integrate repressive DNA elements in proximity to transcription start sites (TSS) of lncRNA genes, including RNA destabilizing elements [20] or polyadenylation cassettes [21]. Unfortunately, this procedure involves time-consuming cloning of HDR donor templates and is hindered by low recombination efficiency of the HDR pathway [22].

In addition to enabling local genetic alterations, the CRISPR/Cas9 system can also be used to generate genomic deletions via simultaneously introducing DSBs at two genomic sites. After CRISPR/Cas9 activity, both ’naked’ DNA ends are fused by end-ligation repair, resulting in deletion of the entire genomic region [23]. This methodology was applied to generate lncRNA-null alleles, either by deleting lncRNA promotors [24] or entire loci [25,26]. However, as lncRNAs frequently intersect with loci encoding protein-coding mRNAs or other noncoding RNAs, minimal interference of the lncRNA transcriptional unit is paramount. To minimize confounding removals of genomic elements such as overlapping genes, promotors/enhancers, epigenetic marks or topologically associating domains (TADs), we propose limiting deletions to lncRNA TSS and exon 1, which, based on our experience, sufficiently abrogates lncRNA expression. Yet, we cannot exclude that certain lncRNA transcripts or splicing isoforms are not affected and we hence recommend testing for the expression of remaining lncRNA exons. Of note, this targeting strategy is only applicable when the deletion is confined to TSS and exon 1 of the lncRNA-of-interest and does not overlap with other transcriptional units or genomic regulatory sequences.

#### 2.1.2. Long Noncoding RNA Target Selection (Step 1)

Using in-house RNA Sequencing (RNA-Seq) data sets, we identified the lncRNA *Gm15441* as a liver-enriched transcript, whose expression is strongly altered upon metabolic disease and in response to short-term calorie restriction (in preparation unpublished data). The genomic locus of lncRNA *Gm15441* is located on murine chromosome 3 (chr3: 96,555,765-96,566,801), yet overlaps with the protein-coding gene *Txnip*, a well-studied modulator of energy metabolism [27,28,29]. For reasons discussed above, we aimed to avoid disruptions of the *Txnip* locus by solely deleting exon 1, which is unique to *Gm15441* (Figure 1a). Using web-based tools such as CRISPR Design [30] and CRISPOR [31], we identified crRNA spacer sequences flanking the first exon of *Gm15441* and selected two 18-nucleotide and two 20-nucleotide spacer sequences, respectively, for each 5′- and 3′-end of the targeted region (Figure 1b). As crRNA spacer binding is critical for successful gene-targeting, we selected crRNA sequences with minimally predicted off-target potential and performed analysis of on-target activity in vitro using cost-effective, self-synthesized sgRNA molecules, which can be produced in high amounts using standard molecular cloning techniques and target-specific polymerase chain reaction (PCR) primer.

#### 2.1.3. Validation of Single Guide RNA Activity In Vitro (Step 2):

Despite advancements in sgRNA prediction algorithms and the availability of user-friendly online tools, in vitro validation of sgRNA activity remains paramount before performing time-consuming and laborious and expensive pronuclear microinjection (PNI). For screening purposes and to ensure maximal in vitro gene-editing, we recommend using cell lines with a high proliferative capacity and transfection efficiency. For our purposes, we used C57BL/6 mice-derived mouse motor neuron-like hybrid cells (NSC-34) [32,33] for transfection of different sgRNA combinations and evaluated gene-editing efficiency via sequence-specific T7 Endonuclease I (T7E1) assays and standard PCR protocols. In T7E1 assays, PCR products containing the gene-targeted region are briefly heated followed by gradual cooling to allow separation of double-stranded DNA homoduplexes, followed by re-annealing of DNA strands. In the case of varying sequence compositions of PCR products due to genome-editing events at the locus of interest, this procedure results in the formation of double-stranded DNA heteroduplexes consisting of unmodified and mutated DNA sequences. As DNA heteroduplexes are a consequence of base pairing mismatches, they can be readily detected as cleaved fragments on an agarose gel when re-annealed PCR products are incubated with T7E1, a mismatch-specific DNA endonuclease that recognizes base-substitution mismatches as well as mismatches resulting from genomic deletions or insertions. To improve detection of gene-editing events, we recommend designing PCR primers in such a way that T7E1-mediated cleavage of heteroduplexes results in two DNA fragments of equal lengths, which gives a strong single band during gel electrophoretic separation (Figure 2a). If both sites flanking the genomic region to be deleted were successfully targeted, the deletion event can also be detected by PCR using primers flanking the excised region. In our approach, which targets exon 1 of *Gm15441*, we observed sgRNA-mediated endonuclease events in six out of eight selected sgRNAs (Appendix A). Furthermore, deletion of the targeted genomic region was confirmed by deletion PCRs when pairs of functional sgRNAs were transfected in combination (Figure 2b).

#### 2.1.4. Single Guide RNA–Cas9 Ribonucleoprotein Assembly and Pronuclear Microinjection (Step 3)

For the generation of lncRNA knockout models, we selected two 20-nucleotide crRNA spacer sequences that demonstrated maximum activity upon in vitro validation and flanked a genomic region of 407 bp (Figure 1b). We then employed PNI of CRISPR/Cas9 components into C57BL/6NRj zygotes, which were subsequently implanted into pseudopregnant RjHan:NMRI females (Figure 3a,b). For maximum quality of injected CRISPR/Cas9 components, synthetic crRNA and tracrRNA molecules were purchased from commercial distributors. Before injection, target-specific crRNAs and sequence-independent tracrRNAs were incubated with recombinant Cas9 proteins to allow assembly of RNP complexes [19]. To enhance the efficacy of CRISPR/Cas9 genome-editing, we further added Cas9-encoding mRNA to the injection mix.

PNI of CRISPR/Cas9 components into zygotes resulted in 106 viable two-cell stage embryos, which were transferred into five pseudopregnant foster females, of which four gave birth to 18 pups overall. Genotyping of founder animal tail biopsies demonstrated multiple successful alterations of the *Gm15441* locus, including truncated alleles in a range of predicted deletion events, but also unpredicted edited alleles with slightly increased or decreased genomic sizes. Interestingly, we also observed four founder animals fully lacking the wildtype *Gm15441* allele, suggesting homozygous CRISPR/Cas9-mediated deletion of *Gm15441* or genomic deletions of primer binding sequences (Figure 3c).

#### 2.1.5. Identification and Validation of Long Noncoding RNA-Deficient Founder Animals (Step 4)

When using CRISPR/Cas9 for generating genome-engineered mouse models, the developmental time point of the genome-editing event and the exact CRISPR/Cas9-induced genomic alteration occur in a random fashion. Thus, the genetic makeup of the targeted locus differs amongst individual founder animals, but also between somatic or gametic cells of the same animal (genetic mosaicism). In order to ensure germline transmission of the edited allele and to establish a new genome-edited mouse line, all animals obtained by CRISPR/Cas9 PNI as well as resulting F1 offspring were investigated for compositions of the target locus by DNA genotyping in somatic tissue biopsies by Sanger sequencing. To this end, we bred all potential founder animals with C57BL6/N wildtype mice and tested for propagation of edited *Gm15441* alleles amongst 2–3 litters obtained per founder. Genotyping of offspring confirmed successful germline transmission of engineered *Gm15441* alleles in four founder animals (termed *Gm15441*^∆1^, *Gm15441*^∆6^, *Gm15441*^∆11^ and *Gm15441*^∆13^). However, not every animal of the positive F1 generations carried edited *Gm15441* alleles (*Gm15441*^wt/∆^ birth rate 17–45%), demonstrating genetic mosaicism within the gametic cells of all positive founders. Intriguingly, one founder animal did not harbor edited alleles of *Gm15441* in genomic tail biopsy DNA (*Gm15441*^∆11^), yet gave rise to *Gm15441*-null animals, which additionally showcases the high degree of mosaicism in gametic and somatic cells in the respective F0 animal. Sanger sequencing in somatic tissues of positive F1 and F2 offspring demonstrated successful on-target editing as well as isogenic compositions of the edited *Gm15441* alleles amongst animals of the respective founder mouse lines, but also revealed variations in the size of genomic deletions (range 411 bp–433 bp), likely due to error-prone NHEJ-repair of Cas9-induced DSBs (Appendix A).

Next, we characterized the edited *Gm15441* allele of *Gm15441*^∆1^, which exhibited the expected 407 bp deletion with additional loss of 13 basepairs at the 5′ Cas9 cleavage site, resulting in a genomic deletion of 420 bp (Figure 4a,b). The quantitative PCR (qPCR) expression analysis in livers of mice harboring wildtype, heterozygous or homozygous compositions of the *Gm15441*^∆1^ allele confirmed the absence of *Gm15441* expression in *Gm15441*^∆/∆^ mice (range 94–97% reduction), yet no discernible alterations in heterozygous *Gm15441*^wt/∆^ knockout mice. *Gm15441*^∆1^ mutant mice exhibited normal fertility and Mendelian inheritance of *Gm15441*^∆1^ alleles (Appendix A). Importantly, the expression of the *Gm15441*-overlapping *Txnip* gene remained unaltered in *Gm15441*^∆/∆^ and *Gm15441*^wt/∆^ livers (Figure 4c).

Taken together, we here demonstrate that the provided protocols can result in the generation of several lncRNA-deficient mouse lines with slightly varying genomic compositions. Contingent on the presence of a lncRNA-selective transcriptional start site, this methodology provides a rapid and cost-effective means to generate lncRNA in vivo mouse models in only 18–20 weeks.

### 2.2. Experimental Design

#### 2.2.1. Animal Use Authorization

All animals were housed in individually ventilated cages (IVC Type II long) in a specific pathogen-free (SPF) research facility with controlled temperature (22–24 °C), relative humidity of 50–70% and a constant, 12-h light/dark cycle. Care of animals was within institutional animal-care committee guidelines approved by local (Bezirksregierung Köln) and regional (Tierschutzkommision acc. §15 TSchG of the Landesamt für Natur, Umwelt und Verbraucherschutz (LANUV) North-Rhine Westphalia, Germany, internal reference No. 84-02.04.2016.A460) authorities. Upon weaning, mice were fed standard rodent chow (Teklad Global Rodent T.2018.R12, Harlan Laboratories, Madison, WI, USA) with ab libitum access to food and drinking water. Before tissue collection, animals were sacrificed by cervical dislocation.

#### 2.2.2. Single-Guide RNA Design

Candidate sgRNAs were designed using CRISPOR [31] to identify crRNA spacer sequences with the highest specificity score (>50) and maximum score by the CRISPRscan algorithm [34]. Predictions were performed using the mouse genome assembly GRCm38/mm10 with the following genome coordinates: chr3:96,566,517-96,567,016.

#### 2.2.3. Single-Guide RNA Synthesis

DNA templates for sgRNA synthesis were generated using 200 ng plasmid pX330-U6-Chimeric_BB-CBh-hSpCas9 (42230, Addgene, Cambridge, MA, USA) and the Phusion^®^ High-Fidelity DNA Polymerase Kit (M0530, NEB, Ipswich, MA, USA) according to the manufacturer’s protocol. Forward primer oligonucleotides were specifically designed for each sgRNA according to the following composition:

T7 promotor-5′-TTAATACGACTCACTATAGG-3′gRNA of interest-variable (18-20 bp of specific gRNA, see Appendix A)pX330-5′tracrRNA-5′-GTTTTAGAGCTAGAAATAGC-3′

For all reactions, generic reverse primer oligonucleotides were used with the following sequence: AAAAAGCACCGACTCGGTGCC. Before RNA synthesis, DNA template PCR reactions were controlled for the appearance of a single specific band (120 or 122 bp) by agarose gel electrophoresis and extracted from agarose gels using the QIAquick^®^ Gel Extraction Kit (28706, Qiagen, Venlo, The Netherlands). sgRNA oligonucleotides were synthesized using 200 ng sgRNA template DNA and the HiScribe™ T7 High Yield RNA Synthesis Kit (E2040, NEB) according to the manufacturer’s protocol. Finally, template DNA was removed from the reaction mix by DNAse I (M0303, NEB) treatment and the synthesized sgRNAs were purified using the NucleoSpin^®^ RNA Kit (740955, Macherey-Nagel, Düren, Germany) according to the manufacturer’s protocol.

#### 2.2.4. In Vitro Single-Guide RNA Validation

Validation of sgRNA activity was performed using NSC-34 cells grown in DMEM GlutaMAX™, high glucose (10569010, Thermo Fisher Scientific, Waltham, MA, USA) supplemented with 10% (*v*/*v*) fetal bovine serum (P30-3301, Pan-Biotech, Aidenbach, Germany), 1% (*v*/*v*) 100X l-Glutamine solution (25030081, Thermo Fisher Scientific) and 1% (*v*/*v*) Penicillin-streptomycin solution (15140122, Thermo Fisher Scientific). Then, 1 × 10^5^ NSC-34 cells per well were seeded on 12-well tissue culture plates (356500, Corning Inc., Corning, NY, USA) and transfected with different combinations of sgRNAs using 500 µL Opti-MEM™ Reduced Serum Medium (31985062, Thermo Fisher Scientific) supplemented with 8 µL/mL Lipofectamine^®^ 2000 (11668019, Thermo Fisher Scientific), 2 µg/mL Cas9 mRNA (L-6125-20, TriLink BioTechnologies, San Diego, CA, USA) and 1 µg/mL of each sgRNA to be tested. To assess sgRNA gene-targeting efficiency, genomic DNA was extracted from NSC-34 cells and used in sgRNA-specific PCR reactions composed of 10X DreamTaq™ Green Buffer (B71, Thermo Fisher Scientific), 10 mM dNTP Mix (R0182, Thermo Fisher Scientific), 1.25 U DreamTaq™ Green DNA Polymerase (EP0711, Thermo Fisher Scientific) and 10 µM sgRNA-specific primer oligonucleotides. All sgRNA-specific primer oligonucleotides were designed in such a way that the resulting PCR product contained the targeted CRISPR/Cas9 cutting site at its center and are listed below. To allow the formation of DNA heteroduplexes, amplified PCR products were heated at 95 °C for 10 min and then incubated at RT for 30 min, before digestion with T7 Endonuclease I (M0302, NEB) at 37 °C for 60 min. Finally, PCR reaction products were analyzed by agarose gel electrophoresis for T7E1-mediated DNA cleavage, which reflects successful sgRNA/Cas9-mediated gene-targeting.


Primer for T7 Endonuclease I Assays


Sequence (5′->3′)
Gm15441-5′-1 (20) forward
GCTCCTACTCAGACCCTTGTTCGm15441-5′-1 (20) reverse
CTCCCTGAGTTGCTTTTGGTCGm15441-5′-2 (20) forward
GAAGGGAGATAAAGCGCACGGm15441-5′-2 (20) reverse
ATGGGGAGCAAGCCGATAAGGm15441-3′-1 (20) forward
GACTAGTCTGATGGAGGCATCGm15441-3′-1 (20) reverse
TGTGTGTGTGTGTGAGAGAGAGGm15441-3′-2 (20) forward
TCAGCCTGCTTTCTTATATGGCGm15441-3′-2 (20) reverse
TGCAAACACAGACATGCACAC


Gm15441-5′-1 (18) forward
GCGCACGTTTAACTGACTCTCGm15441-5′-1 (18) reverse
ATAAGCAGCACCCCTCCATGGm15441-5′-2 (18) forward
CACAGAAGGGAGATAAAGCGCGm15441-5′-2 (18) reverse
TTGCCTTCCCTCACTGATGGGm15441-3′-1 (18) forward
ATCAGTGAGGGAAGGCAAGGGm15441-3′-1 (18) reverse
AGCAAGCCAGTATCACATGCGm15441-3′-2 (18) forward
ATGGAGGGGTGCTGCTTATCGm15441-3′-2 (18) reverse
GCAGGAAGGCTAACAGGAGG

#### 2.2.5. Single-Guide RNA–Cas9 Ribonucleoprotein Assembly and Pronuclear Microinjection

Synthetic guide RNAs were assembled from generic tracrRNAs (1072532, Integrated DNA Technologies Inc., Coralville, IA, USA) and custom crRNAs (custom Alt^®^-R crRNA, Integrated DNA Technologies Inc.) using the following crRNA spacer sequences: GGCCTTGGCTCACTAGGTGA (5′) and TTCCCAGATGACTTTAGTTG (3′). Annealed gRNAs were incubated with Cas9 proteins to obtain functional ribonucleoprotein (RNP) complexes, followed by addition of Cas9 mRNA to the injection mix. The final injection mix contained 400 nM of each gRNA, 200 nM Cas9 protein (1074181, Integrated DNA Technologies Inc.) and 30 ng/µL Cas9 mRNA (L-6125-20, TriLink BioTechnologies) in T10E0.1 buffer (10 mM Tris-HCl, 0.1 mM EDTA, embryo-tested water (W1503, Sigma-Aldrich, St. Louis, MO, USA). Pronuclear microinjections (PNI) of CRISPR/Cas9 components into fertilized oocytes of superovulated C57BL/6NRj females were performed at the CECAD in vivo Research Facility, Cologne, Germany as described in [26]. Healthy, 2-cell stage embryos were subsequently implanted into the oviduct of pseudo-pregnant RjHan:NMRI females.

#### 2.2.6. Genotyping and Gene Expression Analysis

Genomic DNA from small tail biopsies of 3-week-old mice was extracted in 300 µL tail lysis buffer in a thermocycler (56 °C, 300 rpm, overnight), followed by proteinase K inactivation (96° C, 300 rpm, 15 min). The tail lysis buffer contained 50 mM KCl (P9333, Sigma-Aldrich), 10 mM Tris-HCl, pH 8.3 (A3452, AppliChem GmbH, Darmstadt, Germany) 1.5 mM MgCl_2_ (M2393, Sigma-Aldrich), 0.45% (*v*/*v*) Tween^®^ 20 (437082Q, VWR International, Radnor, PA, USA), 0.45% (*v*/*v*) Nonidet™ P-40 (A1694, AppliChem GmbH) and 100 µg/mL proteinase K (03115844001, Roche Diagnostics International AG, Rotkreuz, Switzerland). An amount of 2 µL genomic DNA was subsequently used in genotyping PCRs composed of 10X DreamTaq™ Green Buffer (B71, Thermo Fisher Scientific), 10 mM dNTP Mix (R0182, Thermo Fisher Scientific), 1.25 U DreamTaq™ Green DNA Polymerase (EP0711, Thermo Fisher Scientific) and 10 µM primer oligonucleotides with the following sequences:

(P1) 5′-CTCTAGCTCCCAAAGGCACC-3′

(P2) 5′-ACAGATTCAGGGTTGCAGGC-3′

(P3) 5′-TCTAGAGCCTGGAAAAGCGC-3′.

For gene expression analysis, total RNA was isolated from liver biopsies derived from 16–18-week-old mice via phenol-chloroform extraction using peqGOLD TriFast™ (30-2010, VWR International), trichlormethane (4432.1, Carl Roth GmbH + Co. KG, Karlsruhe, Germany), 2-propanol (I9516, Sigma-Aldrich), absolute ethanol (20821.296, VWR International) and DEPC-H_2_O (95284, Sigma-Aldrich). Isolated RNA was converted into complimentary DNA (cDNA) via the High-Capacity cDNA Reverse Transcription Kit (4368814, Thermo Fisher Scientific) according to the manufacturer’s protocol. Quantitative real-time PCR analysis was performed using 2 µL equimolar cDNA solutions, 10X SYBR™ Select Master Mix (4472903, Thermo Fisher Scientific) and 1 µM primer oligonucleotides with the following sequences: CCTTGCCTTCCCTCACTGAT and GATCAGACCATCCATCCTGG. Statistical differences between genotypes were calculated using unpaired, two-tailed Student’s T-tests (UP2T-TT).

## Figures and Tables

**Figure 1 ncrna-05-00012-f001:**
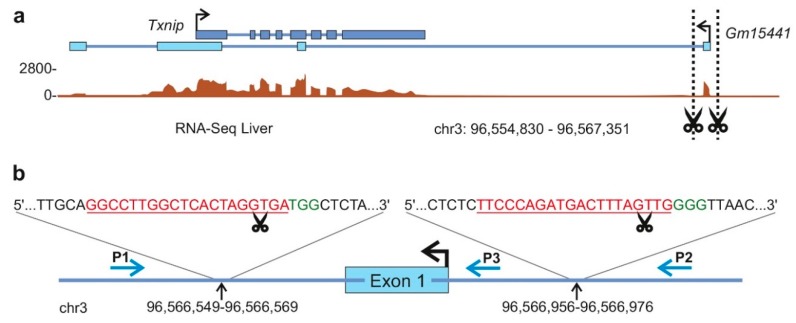
Targeting strategy: (**a**) Schematic illustration of the targeting strategy to delete exon 1 of long noncoding RNA (lncRNA) *Gm15441*. Brown histograms represent RNA-Seq read counts in livers of 30-week-old C57BL/6 mice. Blue boxes indicate annotated exons of lncRNA *Gm15441* (light blue) and overlapping protein-coding gene *Txnip* (dark blue). Dotted lines show genomic sites to be cleaved by clustered regularly interspaced short palindromic repeats (CRISPR)/Cas9-mediated genome engineering. (**b**) Schematic representation of the targeted *Gm15441* locus. Spacer sequences of Cas9-recruiting CRISPR RNAs (crRNAs) are shown in red. Protospacer adjacent motifs necessary for Cas9 activity are shown in green. Black arrows indicate the genomic coordinates bound by crRNA spacer sequences. Scissors depict Cas9 cutting sites. Blue arrows indicate polymerase chain reaction (PCR) primers P1, P2 and P3 used for genotyping and sequencing.

**Figure 2 ncrna-05-00012-f002:**
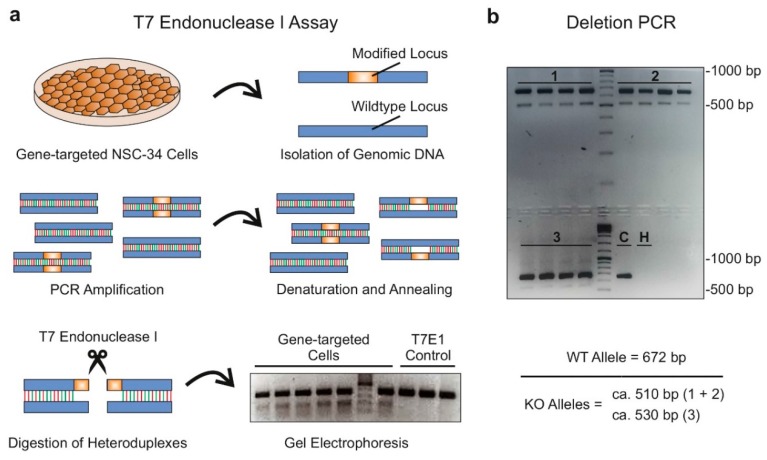
In vitro validation of single guide RNA (sgRNA) activity: (**a**) Schematic representation depicting the workflow of T7 endonuclease I (T7E1) assays. To detect genome-editing events induced by functional sgRNAs, genomic DNA is isolated from gene-targeted cells and the locus of interest is amplified via PCR. After denaturation and random re-annealing of DNA strands, the PCR products are incubated with T7E1, which results in cleavage of DNA heteroduplexes consisting of unmodified and gene-targeted DNA strands. Cleaved heteroduplexes can be detected by gel electrophoresis and demonstrate successful gene-editing by functional sgRNAs. To control for T7E1-specific appearance of additional bands, PCR products that have not been subjected to T7E1 digestion are also run on the same gel electrophoresis (T7E1 control). (**b**) Representative agarose gel displaying PCR reactions with primers flanking exon 1 of lncRNA *Gm15441* using genomic DNA isolated from NSC-34 cells transfected with different combinations of sgRNAs, including [1] *Gm15441-5′-2 (20)* + *Gm15441-3′-2 (18)*, [2] *Gm15441-5′-1 (18)* + *Gm15441-3′-2 (18)* and [3] *Gm15441-5′-2 (18)* + *Gm15441-3′-2 (18)*. C, negative control PCR using genomic DNA from untransfected NSC-34 cells. H, H_2_O control PCR using no template DNA.

**Figure 3 ncrna-05-00012-f003:**
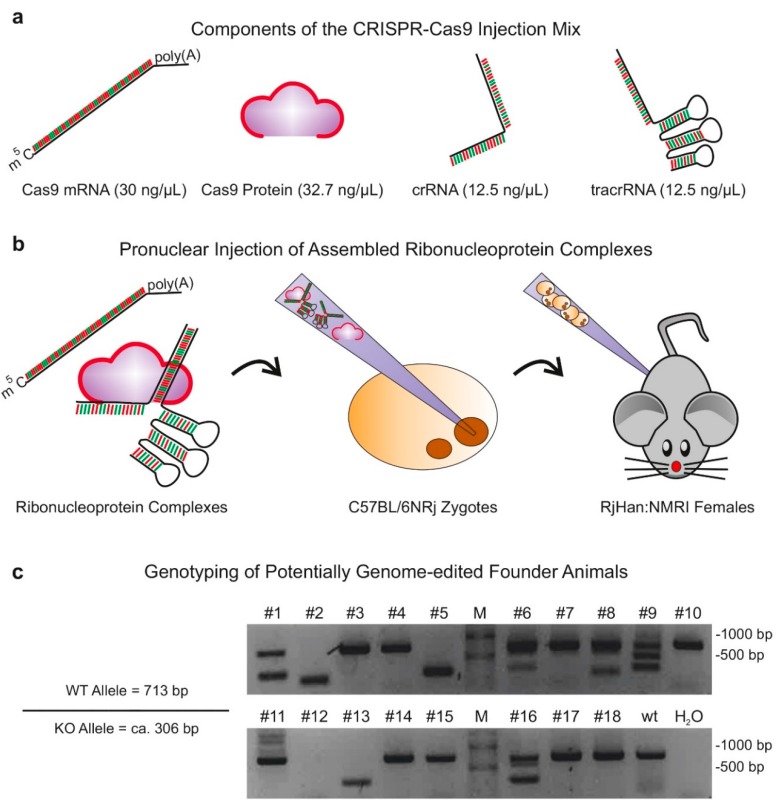
In vivo gene-targeting: Schematic representation of the experimental approach to generate in vivo mouse models, depicting the (**a**) composition of the CRISPR/Cas9 injection mix, containing Cas9 mRNA, Cas9 protein as well as the gRNA components trans-activating CRISPR RNA (tracrRNA) and crRNA; (**b**) pronuclear microinjection of assembled ribonucleoprotein complexes into C57BL/6NRj zygotes and embryo transfer of 2-cell stage embryos into pseudopregnant RjHan:NMRI females. Concentrations of the injection components are indicated in the panel. (**c**) Agarose gel depicting genotyping PCR reactions with primers flanking the targeted genomic region using genomic DNA isolated from 18 individual founder mice, which have been obtained by pronuclear microinjection of CRISPR/Cas9 components. M, marker; wt C57BL/6 wildtype control mouse; H_2_O, no template control PCR.

**Figure 4 ncrna-05-00012-f004:**
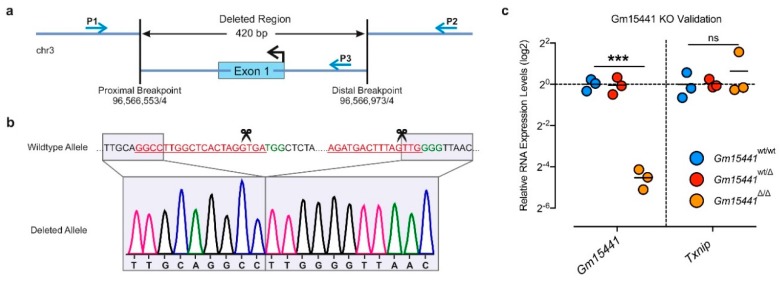
Identification of stable lncRNA-deficient mouse lines: (**a**) Scheme of the genomic locus of founder #1, depicting the *Gm15441*^∆^ allele with proximal and distal breakpoints. Blue arrows indicate PCR primers P1, P2 and P3 used for genotyping and sequencing. (**b**) Chromatogram showing the genomic DNA sequence of the *Gm15441*^∆^ allele in comparison to the wildtype allele. (**c**) qPCR gene expression analysis of lncRNA *Gm15441* and overlapping protein-coding gene *Txnip* in mice harboring wildtype (*Gm15441*^wt/wt^), heterozygous (*Gm15441*^wt/∆^) or homozygous (*Gm15441*^∆/∆^) compositions of the *Gm15441*^∆^ allele (*n* = 3). Graphs represent mean expression values with all data points shown. Statistical differences were calculated using unpaired two-tailed t-tests (UP2T-TT). *** *p* < 0.001; ns, not significant.

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
