# Peer review of "Rapid Generation of Long Noncoding RNA Knockout Mice Using CRISPR/Cas9 Technology"

_ncrna, 2019, doi:10.3390/ncrna5010012_

Round 1
Reviewer 1 Report
Review of submission ncrna-395654: Rapid generation of long noncoding RNA knockout mice using CRISPR/Cas9 technology
The manuscript by Nils R. Hansmeier and colleagues describes nicely how CRISPR/Cas9 can be used to generate and validate lncRNA loss-of-function mouse models. The data is presented in a concise and clear manner, but there are a few points that need revisiting before this manuscript should be accepted for publication.
Major points
1. The Supplementary Figures do not represent what is described in the text or the Figure legends and seem to be mixed up!!!
Supplementary Figure 1 should show a Table of sgRNAs that have been validated and the image is definitely not showing the described table.
Supplementary Figure 2 is unclear as the main text corresponding to Figure S2 and the Figure legend seem to match to the image shown in Figure S1. Also, the internal classifications of the lines (e.g. Line 837) do not make it very clear to which of these founder animals they correspond to (e.g Gm15441Δ1).
Supplementary Figure 3 is showing T7 Endonuclease Assay primers but I am assuming should show the table that is currently in Figure S2? There is no such thing as 9.66 pups so please change or eliminate this column.
2. In line 202 they mention they bred all potential founders. A table clarifying which F0 animals were bred and what genotype the resulting offspring showed would be helpful.
3. Row 180 to 183: they describe that ‘four founder animals are fully lacking the wild type Gm15441 allele’ and conclude that this is either due to homozygous deletion or loss of primer binding sites. From the gel image in Figure 3c it appears that there are 3 mice with homozygous deletions and one mouse that might have lost the primer binding sites. However, the reason for a complete absence in bands in mouse #12 could be due to any variety of different options: there might be a deletion present that is bigger than the expected deletion allele, the 5’ and/or 3’ gRNAs created an indel that deleted the primer binding sequences or there is another rearrangement (inversion/duplication, partial rearrangement) present. It would be interesting to see which, if any, of these potential alleles is present in mouse #12. For example, for larger than expected deletion, one could use primers that are further away from the expected cut sites. For indels deleting primer binding sequence one could amplify both the 5’ and 3’ region and then analyse the sequence.
Also, adding or changing the sentence to make it clear that there were 3 mice which looked like carrying a potential homozygous deletion, of which one (#13) went GLT. Did this mouse have, as expected, all heterozygous offspring? And then one mouse (#12) that did not show a product on the gel and had an allele present as discovered/described with the above analysis.
4. If one generates a cut on both the 5’ and 3’ side of a genomic region, deletions are not the only rearrangement that can be found. Additionally, as is mentioned in the manuscript the F0 founders show genetic mosaicism so a combination of these rearrangements might be present within a single founder. There have been several publications showing that inversions, duplications and partial rearrangements happen, that they can be present within the same founder animal and that they get passed on to their offspring efficiently.
Besides the issue already mentioned in point 2 regarding mouse #12, a potentially underlying additional rearrangement might explain the unexpected normal expression levels in heterozygous Gm15441wt/Δ KO mice in Figure 4c. Thorough analysis of both the founder (F0) as well as the offspring used for generating this line is required to determine if they are clean deletion lines or carry another, underlying allele.
Additionally, the same analysis would be beneficial for the remaining 17 founder animals and their offspring to see whether they do indeed carry clean deletion alleles.
Minor points
1. In the text starting at row 176 a referral to the % birth rate would be helpful.
Author Response
Point 1:
The Supplementary Figures do not represent what is described in the text or the Figure legends and seem to be mixed up!!!
Supplementary Figure 1 should show a Table of sgRNAs that have been validated and the image is definitely not showing the described table.
Supplementary Figure 2 is unclear as the main text corresponding to Figure S2 and the Figure legend seem to match to the image shown in Figure S1. Also, the internal classifications of the lines (e.g. Line 837) do not make it very clear to which of these founder animals they correspond to (e.g. Gm154411∆).
Supplementary Figure 3 is showing T7 Endonuclease Assay primers but I am assuming should show the table that is currently in Figure S2? There is no such thing as 9.66 pups so please change or eliminate this column.
-
It seems the Supplementary Figures have been swapped in the manuscript, which has been sent to Reviewer #1. We double-checked this issue in our manuscript, but couldn’t find any problems there.
However, we agree with Reviewer #1 in that our internal classifications of the lines are unclear and changed the labeling of the mouse lines to ‚Gm15441∆1’ (founder #1), ‚Gm15441∆6’ (founder #6), ‚Gm15441∆11’ (founder #11) and ‚Gm15441∆13’ (founder #13) in all affected Supplementary Figures.
Additionally, we eliminated the column of the mean number of offspring per breeding in New Figure S3.
*Previous version (figure legend for Figure S2):
Overview of Ensemble genome browser tracks depicting Gm15441∆alleles generated by CRISPR/Cas9-mediated deletion of Gm15441 exon 1. Error-prone NHEJ repair of Cas9-induced DSBs resulted in minor variations of truncated Gm15441∆ alleles in the four stable Gm15441∆mouse lines (internal classifications as Line 837, Line 842, Line 847 and Line 849). The deletion size for the respective mouse lines is as follows: Line 837, 420 bp; Line 842, 422 bp; Line 847, 433 bp; Line 849, 411 bp.
-
*Revised version (changes underlined):
Overview of Ensemble genome browser tracks depicting Gm15441∆alleles generated by CRISPR/Cas9-mediated deletion of Gm15441 exon 1. Error-prone NHEJ repair of Cas9-induced DSBs resulted in minor variations of truncated Gm15441∆ alleles in the four stable Gm15441∆ mouse lines (internal classifications as Gm15441∆1[founder #1], Gm15441∆6[founder #6], Gm15441∆11[founder #11] and Gm15441∆13[founder #13]). The deletion size for the respective mouse lines is as follows: Gm15441∆1, 420 bp; Gm15441∆6, 422 bp; Gm15441∆11, 433 bp; Gm15441∆13, 411 bp. All mice of line Gm15441∆13harbor an additional T->A base pair substitution at the 3’ cutting site (chr3: 96,566,985).
Point 2:
In line 202 they mention they bred all potential founders. A table clarifying which F0 animals were bred and what genotype the resulting offspring showed would be helpful.
-
As all F0 animals were bred with wild type C57BL6/N mice, resulting offsprings consequently carried one wild type allele of Gm15441and one Gm15441allele by the potential founders. We think that a table of all breedings, although possible, doesn’t seem to add new information to the story.
However, we edited the text of the manuscript to highlight that all potential founders were bred with C57BL/6N wild type mice, indicating that resulting F1 offspring always carried a wild type allele of Gm15441and maximally one edited Gm15441allele.
*Previous version (line 178-179):
To this end, we bred all potential founder animals and tested for propagation of edited Gm15441alleles amongst 2-3 litters obtained per founder.
-
*Revised version (changes underlined):
To this end, we bred all potential founder animals with C57Bl6/N wild type miceand tested for propagation of edited Gm15441alleles amongst 2-3 litters obtained per founder.
Point 3:
Row 180 to 183: they describe that ‘four founder animals are fully lacking the wild type Gm15441allele’ and conclude that this is either due to homozygous deletion or loss of primer binding sites. From the gel image in Figure 3c it appears that there are 3 mice with homozygous deletion and one mouse that might have lost the primer binding sites. However, the reason for a complete absence in bands in mouse #12 could be due to any variety of different options: there might be a deletion present that is bigger than the expected deletion allele, the 5’ and/or 3’ gRNAs created an indel that deleted the primer binding sequences or there is another rearrangement (inversion/duplication, partial rearrangement) present. It would be interesting to see which, if any, of these potential alleles is present in mouse #12. For example, for larger than expected deletion, one could use primers that are further away from the expected cut sites. For indels deleting primer binding sequence one could amplify both the 5’ and 3’ region and then analyse the sequence.
Also, adding or changing the sentence to make it clear that there were 3 mice which looked like carrying a potential homozygous deletion, of which one (#13) went GLT. Did this mouse have, as expected, all heterozygous offspring? And then one mouse (#12) that did not show a product on the gel and had an allele present as discovered/described with the above analysis.
-
We agree with Reviewer #1 in that there are many possible genomic rearrangements that could cause the complete absence of bands on the gel image for founder mouse #12. Unfortunately, we never sequenced the Gm15441locus of founder mouse #12 and had to sacrifice this founder due to a penis prolaps in december 2017. By now, there are no samples of founder #12 left and we don’t have the possibility to check the composition of its Gm15441locus anymore. However, although theoretically very interesting, the detailed compositions of all generated Gm15441loci has not been the main focus of our work and doesn’t add extra value to our technical protocol.
The second sub-point raised by Reviewer #1 is actually very interesting, as not all offspring by founder mouse #13 (as well as of the other positive founders) were heterozygous for the Gm15441locus, but predominantly homozygous for wild type Gm15441alleles. We thus added an extra sentence to emphasize on the sporadical propagation of edited Gm15441alleles by all four positive founders.
*Previous version (line 179-184):
Genotyping of offsprings confirmed successful germline transmission of engineered Gm15441 alleles in four founder animals (termed Gm15441∆1,Gm15441∆6, Gm15441∆11 and Gm15441∆13). Intriguingly, one founder animal that did not harbor edited alleles of Gm15441in genomic tail biopsy DNA (Gm15441∆11), yet gave rise to Gm15441-null animals, which showcases the high degree of mosaicism in gametic and somatic cells in the respective F0 animal.
-
*Revised version (changes underlined):
Genotyping of offsprings confirmed successful germline transmission of engineered Gm15441 alleles in four founder animals (termed Gm15441∆1,Gm15441∆6, Gm15441∆11 and Gm15441∆13).However, not every animal of the positive F1 generations carried edited Gm15441alleles (Gm15441wt/∆birth rate 17 % - 44 %), demonstrating genetic mosaicism within gametic cells of all positive founders.Intriguingly, one founder animal that did not harbor edited alleles of Gm15441in genomic tail biopsy DNA (Gm15441∆11), yet gave rise to Gm15441-null animals, which additionallyshowcases the high degree of mosaicism in gametic and somatic cells in the respective F0 animal.
Point 4:
If one generates a cut on both the 5’ and 3’ side of a genomic region, deletions are not the only rearrangement that can be found. Additionally, as is mentioned in the manuscript the F0 founders show genetic mosaicism so a combination of these rearrangements might be present within a single founder. There have been several publications showing that inversions, duplications and partial rearrangements happen, that they can be present within the same founder animal and that they get passed on to their offspring efficiently.
Besides the issue already mentioned in point 2 regarding mouse #12, a potentially underlying additional rearrangement might explain the unexpected normal expression levels in heterozygous Gm15441wt/∆ KO mice in Figure 4c. Thorough analysis of both the founder (F0) as well as the offspring used for generating this line is required to determine if they are clean deletion lines or carry another, underlying allele.
Additionally, the same analysis would be beneficial for the remaining 17 founder animals and their offspring to see whether they do indeed carry clean deletion alleles.
-
The Gm15441alleles of all positive F1 and F2 animals of all mouse lines have been sequenced to validate identical compositions of the edited Gm15441loci among siblings and succeeding offspring. Indeed, Sanger sequencing confirmed identical compositions of the editedGm15441alleles for all positive founder lines. However, the manuscript didn’t clearly address this legitimate issue and hence was changed to highlight the isogenic configuration within the positive mouse lines.
*Previous version (line 184-186):
Sanger sequencing in somatic tissues of all founders demonstrated successful on-target editing, but also revealed variations of size of genomic deletions (range 411 bp - 433 bp), likely due to error-prone NHEJ-repair of Cas9-induced DSBs (Figure S2).
-
*Revised version (changes underlined):
Sanger sequencing in somatic tissues of positive F1 and F2 offspringdemonstrated successful on-target editing as well as isogenic compositions of the edited Gm15441alleles amongst animals of the respective founder mouse lines, but also revealed variations of size of genomic deletions (range 411 bp - 433 bp), likely due to error-prone NHEJ-repair of Cas9-induced DSBs (Figure S2).
Minor points
Point 1:
In the text starting at row 176 a referral to the % birth rate would be helpful.
-
The birth rate of positive Gm15441wt/∆F1 offspring is now included in the extra sentence addressing Major Point #3 by Reviewer #1.

Reviewer 2 Report
The authors present an advanced methodology method in generating lncRNA and further suggest the application of CRISPR/Cas9 for lncRNA genes. This study is well designed and the results go straightforwardly.
However, I have some comments for this manuscript.
Major points:
1. The authors mentioned about a liver-specific lncRNA Gm15441 in Abstract and did not mentioned more in Introduction part, even though, whole paper the authors almost worked with this lncRNA.
2. Broadly, it will be more logical and attractive to the readers if the authors can explain why did they choose lnc Gm15441 for their proof-of-principle approach.
3. The authors suggested that, this methodology is an advanced method with rapid and cost-effective mean.
It may also be ambiguous to the readers. Is there any standard comparison to make this conclusion?
Minor point(s):
Reference style of this manuscript did not follow the submitted journal well.
Author Response
Point 1:
The authors mentioned about a liver-specific lncRNA Gm15441 in Abstract and did not mentioned more in Introduction part, even though, whole paper the authors almost worked with this lncRNA.
-
The main focus of this publication is the methodology to generate lncRNA loss-of-function mouse models and thus we decided not to distract from the technical procedure or to share unpublished findings of an ongoing research project, which I hope the reviewer will understand.
However, the specification ‚liver-specific lncRNA Gm15441‘ in the abstract is unwarranted and thus was changed to ‚liver-enriched lncRNA Gm15441‘.
*Previous version (line 31-33):
In a proof-of-principle approach, we generated mice deficient for the liver-specific lncRNA Gm15441, which we found downregulated during development of metabolic disease and induced during the feeding/fasting transition.
-
*Revised version (changes underlined):
In a proof-of-principle approach, we generated mice deficient for the liver-enrichedlncRNA Gm15441, which we found downregulated during development of metabolic disease and induced during the feeding/fasting transition.
Point 2:
Broadly, it will be more logical and attractive to the readers if the authors can explain why did they choose lnc Gm15441 for their proof-of-principle approach.
-
As this lncRNA is part of an ongoing project (unpublished data) and hitherto mostly uncharacterized, we prefer not to mention more details about the biological significance or the underlying disease pathogenesis associated with this lncRNA. However, we briefly reasoned why we investigate Gm15441in Step 1 of our methodology and consider this appropriate for a technical publication.
*Previous version (line 112-114):
‘Using in-house RNA sequencing (RNA-Seq) data sets, we identified the lncRNA Gm15441 as liver-enriched transcript, whose expression is strongly altered upon metabolic disease and in response to short-term calorie restriction (unpublished findings).’
Point 3:
The authors suggested that, this methodology is an advanced method with rapid and cost-effective mean.
It may also be ambiguous to the readers. Is there any standard comparison to make this conclusion?
-
However, to better elucidate the rapid and cost-effective mean of our procedure, we added another sentence to the introduction illustrating the benefits of our methodology in comparison to conventional gene targeting.
*Revised version (changes underlined):
Classical gene targeting aproaches using embryonic stem (ES) cells involve cloning of the targeting construct, rare homologous recombination (HR) events in ES cells and extensive ES cell culture techniques, for instance ES cell electroporation, expansion and selection. These procedures can take up to several months to generate a correctly targeted ES cell clone and ultimately a homozygous transgenic mouse line, whereas our methodology can generate loss-of-function mouse models within 18-20 weeks.
Minor points
Point 1:
Reference style of this manuscript did not follow the submitted journal well.
-
The reference style of our manuscript has been adapted to MDPI guidelines.
